# Cytotoxic Properties of Polyurethane Foams for Biomedical Applications as a Function of Isocyanate Index

**DOI:** 10.3390/polym15122754

**Published:** 2023-06-20

**Authors:** Dominik Grzęda, Grzegorz Węgrzyk, Adriana Nowak, Joanna Idaszek, Leonard Szczepkowski, Joanna Ryszkowska

**Affiliations:** 1Faculty of Materials Science and Engineering, Warsaw University of Technology, Wołoska 141, 02-507 Warsaw, Poland; grzegorz.wegrzyk.dokt@pw.edu.pl (G.W.); joanna.idaszek@pw.edu.pl (J.I.); joanna.ryszkowska@pw.edu.pl (J.R.); 2Department of Environmental Biotechnology, Lodz University of Technology, Wolczanska 171/173, 90-530 Lodz, Poland; adriana.nowak@p.lodz.pl; 3Fampur Adam Przekurat, 85-825 Bydgoszcz, Poland; leonardosz@interia.pl

**Keywords:** polyurethane, foams, composite matrix, cytotoxicity, biomedical, open-cell

## Abstract

Polyurethane foams are widely used in biomedical applications due to their desirable mechanical properties and biocompatibility. However, the cytotoxicity of its raw materials can limit their use in certain applications. In this study, a group of open-cell polyurethane foams were investigated for their cytotoxic properties as a function of the isocyanate index, a critical parameter in the synthesis of polyurethanes. The foams were synthesized using a variety of isocyanate indices and characterized for their chemical structure and cytotoxicity. This study indicates that the isocyanate index highly influences the chemical structure of polyurethane foams, also causing changes in cytotoxicity. These findings have important implications for designing and using polyurethane foams as composite matrices in biomedical applications, as careful consideration of the isocyanate index is necessary to ensure biocompatibility.

## 1. Introduction

Biostable polyurethanes (PU) have been used for many years to manufacture long-lasting medical implants [1,2,3]. The development of PU has led to the development of many components used in medicine, such as bone fixation [4], coatings [5], artificial hearts [6], heart valves, and aortic prostheses [7]. Biomedical applications currently include catheters, stents, heart valves, dialysis devices, dressings, adhesives, drug delivery devices, pacemakers, and breast implant shells [8,9].

Such widespread PU applications in medicine were due to their mechanical flexibility, high tear strength, biocompatibility, and abrasion resistance. The popularization of PU applications in medicine was limited by the toxicity of 4,4-methylene diphenyl diisocyanate (MDI) [10]. Consequently, PU made with MDI has found applications in rigid and flexible foams, coatings, adhesives, sealants, and elastomers. [11]. The variety of products, which determines the wide range of applications, is due to the variability of PU formulations [12,13]. The quantities and types of polyols, isocyanates, catalysts, and other additives used in their synthesis can be varied in the formulations. There is a trend on the market to develop new PU products with improved properties over existing ones, from conventional polyether polyols, polyester polyols, polycarbonate diol, and MDI. Among PU applications in medicine, polyurethane elastomers dominate, and a lot of research has been devoted to these materials [4,5,6,7,8,9]. Polyurethane elastomers are produced with an equal molar proportion of substrates.

In recent years, flexible polyurethane foams have been increasingly used in biomedical applications. As in the case of polyurethane elastomers, the properties of flexible PU foams can be shaped by selecting the type and proportion of substrates for the manufacture of these materials [14,15]. These materials are used to make bandages, mattresses, auxiliary materials used in orthopedics, and insoles for shoes.

An important application of PU is products to prevent pressure sores. Bedsores are ischemic skin and underlying tissue wounds caused by continuous pressure from friction or shear between the external surface and bone or cartilage. Prolonged pressure can reduce blood flow in capillaries and lead to cell death, necrosis, ruptured tissue, or additional serious complications, including osteomyelitis, septicemia and contractures, atrophy, and psychiatric disorders [16,17,18]. Similar symptoms were observed in patients using prosthetic limbs.

The different types of PU have been extensively studied [19,20,21,22], but few studies evaluate the effect of PU chemical structure on the cytotoxicity of foams [23].

The literature most often analyzes the effect of various modifying additives on changing the characteristics of foams made with the selected formulation. Because of their excellent antibacterial activity, gold nanoparticles or silver nanoparticles (AgNPs) are being used to modify flexible foams [24,25,26].

An interesting group of hybrid organic-inorganic materials is polyhedral oligomeric silsesquioxanes, POSS [27], and their homo derivates.

In the case of viscoelastic foams, this is quite a non-issue, given their main applications, such as mattresses and pillows. In such applications, the human body is constantly exposed to the possible adverse effects of their influence.

Studying the impact of individual synthesis components will allow for targeting and further development of these foam applications.

Previous studies have examined the effect of using different catalysts necessary for the synthesis of polyurethane foams on their properties [28]. It turned out that different sets of catalysts affect the mechanical properties of foams and, most importantly, their cytotoxicity. 

Besides the type of substrates used, the main factor influencing the characteristics of foams is the isocyanate index, determined by the molar ratio of isocyanate groups to hydroxide groups.
(1)$$INCO=\frac{\left(n+m\right)NCO}{nOH+mNH_{2}}$$

The hydroxyl groups react with isocyanate groups to form carbamic acid esters. The term “urethanes” is derived from ethyl carbamate.




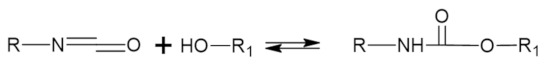

(2)



The number of amine groups is included in the equation, because amine compounds with isocyanates form asymmetric ureas. Amines catalyze the reaction of isocyanates with water and alcohol.




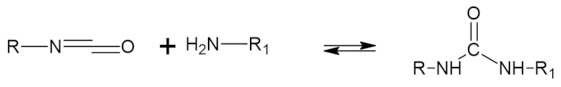

(3)



Water also needs to be taken into account in the equation. It reacts with isocyanates, forming an unstable carbamic acid compound that breaks down into a primary amine and carbon dioxide.




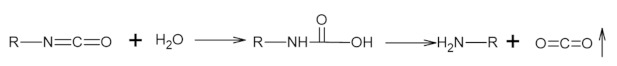

(4)



The research subjects were flexible open-cell polyurethane foams with good air permeability, ultimately acting as a composite matrix. These foams are usually produced with an underabundance of isocyanate groups in the reaction medium.

## 2. Materials and Methods

### 2.1. Materials

Component A of the polyols Voranol CP 1421, Rokopol D2002, and Rokopol G500, were used to produce the foams, the contribution of the individual polyols being the know-how of the authors of the article. Two catalysts, Dabco NE 1070 and Jeffcat DPA, and two surfactants, Tegostab B8404 and Tegostab B 4900, were also introduced to the A component. Ongronat TR 4040 was used as component B.

A description of the components used for the synthesis of PUFs is as follows:Polyols: Voranol CP 1421, polyether triol with a high content of ethylene oxide (33 mg KOH/g, Average Molecular Weight: 5023 g/mol, Dow, MI, USA), Rokopol D2002—a polyoxypropylenediol (53–59 mg KOH/g, Average Molecular Weight: 2000 g/mol, PCC Rokita, Brzeg Dolny, Poland) and Rokopol G500—a polyoxyalkylenetriol (290–310 mg KOH/g Average Molecular Weight ~560 g/mol, PCC Rokita, Brzeg Dolny, Poland);Catalysts: Dabco NE 1070-N, N-dimethyl aminopropyl urea, (H_2_NCO)NH(CH_2_)_3_N(CH_3_)_2_, molecular weight 145 g/mol (Air Products, Allentown, Pennsylvania, USA); trifunctional, amine, reactive catalyst, and Jeffcat DPA (Huntsman, The Woodlands, TX, USA)-N-(3-dimethylamino-propyl)-N,N-diisopropanolamine, amine(III-terminal amine), reactive-, two hydroxyl groups catalyst;Surfactants: silicone surfactants Tegostab B404 and Tegostab B4900 (Evonik Industries, Essen, Germany);Isocyanate: Ongronat TR 4040 isocyanate—a mixture of MDI mixed isomers and oligomeric MDI (31.6–33.6% NCO/wt.%, BorsodChem, Kazincbarcika, Hungary).

As the porophore, 3.1 php (parts per hundred of polyol) of distilled water was used. Foams were made using different isocyanate indices; a set is described in Table 1, their quantitative contribution being the authors’ know-how.

### 2.2. Methods

#### 2.2.1. Synthesis Parameters and Apparent Density

To determine the synthesis parameters: start time, rise time, and gel time, an electronic stopwatch was used. The stopwatch accuracy was 1 s. Start time is defined by the period elapsed from mixing components A and B to the start of the foam growth. Rise time is measured until the foam reaches its maximum height, and the gel time is set when pulling the strand of the polymer with a rod is possible. The apparent density measurements were carried out according to EN ISO 845 [29]. WPA 180/C/1 analytical balance (Radwag, Radom, Poland) was used to determine the weight of the samples with an accuracy of up to ±0.1 mg. Measurements of the 50 × 50 × 50 mm cubes had an accuracy of up to ±0.1 mm.

#### 2.2.2. Fourier Transform Infrared Spectroscopy (FTIR)

The absorption spectra of the foams were examined using a Nicolet 6700 spectrophotometer (Thermo Fisher Scientific Inc., Waltham, MA, USA) with an ATR module to determine their chemical composition. Each sample was scanned 64 times in the 4000–400 cm^−1^. The results were then analyzed using Omnic Spectra 8.2.0 software (Thermo Fisher Scientific Inc., Waltham, MA, USA). For each conventional PUF foam, three spectra were produced.

#### 2.2.3. Foam Extracts Preparation

To test the cytotoxicity, the extracts were prepared according to EN ISO 10993-12:2012 [30]. A total of 2.0 ± 0.05 g of each foam were weighed and suspended in 40 mL of complete culture Dulbecco’s Modified Eagle’s Medium (DMEM) (Merck Life Science, Warsaw, Poland), mixed, and then extracted for 8 h (300 rpm) at 37 ± 1 °C. Each extract was sterile filtered through 0.45 µm and then through 0.22 µm syringe filters (Labindex S.A., Warsaw, Poland).

#### 2.2.4. HaCaT Cell Culture 

The normal immortalized human keratinocyte cell line HaCaT (the original material created by Prof. Dr. Petra Boukamp and Dr. Norbert Fusenig [31] was used in this study. Cells were purchased in Cell Line Service GmbH, Eppelheim, Germany, from the 35th passage. They were cultured as a monolayer in DMEM with the addition of 10% fetal bovine serum (FBS, Gibco, Thermo Fisher Scientific Inc., Waltham, MA, USA), 2 mM glutaMAX^TM^ (Gibco, Thermo Fisher Scientific Inc., Waltham, MA, USA), 25 mM HEPES (Merck Life Science, Warsaw, Poland), 100 μg/mL streptomycin and 100 IU/mL penicillin mixture (Merck Life Science, Warsaw, Poland) for 3–5 days at 37 °C in 5% CO_2_ atmosphere in Galaxy 48S incubator (New Brunswick, UK). After reaching 80% confluence, cells were detached with TrypLE^TM^ Express (Gibco, Thermo Fisher Scientific Inc., Waltham, MA, USA) according to the manufacturer’s instructions, centrifuged (182× *g*, 3 min), decanted, and then fresh DMEM was added. Viability was checked with a trypan blue (Merck Life Science, Warsaw, Poland) exclusion test. The viability of cells taken to each experiment was 90–95%.

#### 2.2.5. Neutral Red Uptake (NRU) Assay

Cytotoxicity was tested with NRU assay according to ISO 10993-5 protocol [32]. The cells were seeded into transparent flat-bottom 96-well plates at 10,000/well in a complete culture medium and were incubated for 24 h at 37 °C in 5% CO_2_. Next, the culture medium was aspirated, and the following concentrations (in four replicates per experiment) of the test extracts were added: 1.56; 3.125; 6.25; 12.5; 25; 50, and 100%. Three independent experiments were conducted. Cells in the culture medium served as a vehicle, whereas cells incubated with dimethylsulfoxide served as a positive control (DMSO, Merck Life Science, Warsaw, Poland) at concentrations from 0.156 to 10%. The assay was conducted for 24 h. After that time, the tested samples were aspirated, and NR (Merck Life Science, Warsaw, Poland) was added in phosphate buffer saline (PBS, Merck Life Science, Warsaw, Poland) in a 50 μg/mL concentration, and the plates were incubated for a further 3 h. The NR solution was gently aspirated and extracted from cells with freshly prepared desorbing solution (1% acetic acid, 50% ethanol, and 49% distilled water). The absorbance was measured at 550 nm using a 620 nm reference filter in a microplate reader TriStar2 LB 942 (Berthold Technologies GmbH and Co. KG, Bad Wildbad, Germany). The absorbance of the control sample (untreated cells) represented 100% cell viability. Cell viability (%) was calculated as (sample OD/control OD) × 100%, and cytotoxicity (%) as 100 − cell viability (%). Results were presented as mean ± standard deviation (SD). The mean error of the method is up to 5–7%. IC_50_ values and non-toxic/safe doses/concentrations (IC_0_) of polyurethane foam extracts were determined from the curves achieved. Morphological changes in HaCaT cells were estimated microscopically using an inverted Nikon Ts2 with EMBOSS contrast (Nikon, Tokyo, Japan) and the Jenoptik Subra Full HD Color under a total magnification of 100×. Qualitative microscopic analysis was conducted, rating the grade from 0 to 4 depending on the reactivity of the cells toward the samples.

## 3. Results

### 3.1. Synthesis Parameters and Apparent Density

Different INCOs alter the synthesis process’s course, changing the start, rise, and gelation times. The measurements of the synthesis parameters are summarized in Table 2.

The timing parameters measured during foam synthesis will be used to assess the calculation of how long the foam should be in the mold in order to correctly complete its synthesis. They also give information on how much time the operator, in the case of non-automated lines, has to close the mold before the synthesis process starts. In the case of this foam system, the foam start time was unchanged at 10 s. Growth and gelation times decreased as the proportion of isocyanate groups in the mixture increased. Increasing INCO resulted in lower apparent density, which is due to the post-reaction of more CO_2_ in the reaction of the isocyanate groups with water.

### 3.2. Fourier Transform Infrared Spectroscopy (FTIR)

Foam syntheses have produced materials with bands characteristic of polyurethanes.

The spectra were merged in Figure 1.

The FTIR spectra of the examined foams exhibit similarity. Specifically, the bands observed in the range of 3600–3400 cm^−1^ in the FTIR spectra of the foams originate from the stretching vibrations of the -OH groups of the polyol hydroxy groups. It can be deduced from the distribution of the multiple bands in this range that PUF_85 contains the most unbonded OH groups, which is twice the amount found in PUF_100 and four times more than PUF_105. If we use a lower INCO during the synthesis of polyurethanes, then there are fewer NCO groups in the reaction mixture. This results in OH groups not having the opportunity to form urethane bonds and remain unbound.

The content of OH groups is due to the INCO and the water content absorbed by the foams during their seasoning. In the case of elastomers made with different INCOs, the average molecular weight (MW) varies; the theoretical relationship between INCO and MW is shown in Figure 2 [33]. At INCOs below 1, polyurethane macromolecules with OH groups are obtained, while at INCOs above 1, they are terminated with NCO groups. The changes in foams MW due to the change in INCO can be expected to be similar. Therefore, PUF_85 contains the most OH groups, and PUF_105 the least with the highest INCO.

Thus, a broad-ranging peak is observable in the 3400–3200 cm−1 range, resulting from both asymmetrical and symmetrical stretching vibrations of the -N-H group present in the urethane groups, urea derivatives, and/or the rest of the catalysts. The bands suggest the deformation vibrations of this group with the maximum at the wavenumber value of 1540 cm^−1^ and 1513 cm^−1^. The -CH group from the CH_3_ and CH_2_ groups exhibit asymmetrical and symmetrical stretching vibrations, resulting in bands at 2971 cm^−1^ and 2868 cm^−1^. Likewise, asymmetrical and symmetrical deformation angles result in a band with a maximum of 1453 cm^−1^ (CH_3_) and 1373 cm^−1^ (CH_2_), and a band with a maximum of 1306 cm^−1^ originating from the stretching vibrations of these groups. The presence of carbonyl groups in the urethane and urea groups is confirmed by multiple signals within the 1750–1640 cm^−1^ range. The stretching vibrations of the C=C group in the aromatic ring result in signal values of 1597 cm^−1^, while the band at the wavenumber of 1227 cm^−1^ comes from the stretching vibrations of the C-N group. The signal originating from the maximum at the wavenumber of 1088 cm^−1^ arises from the C-O stretching that form elastic polyurethane segments.

The formation of urea, allophanate, biuret, and isocyanurate groups occurs through secondary reactions in addition to the primary reaction, which leads to the formation of urethane groups. These groups are also present in polyurethane foams, and their presence was analyzed in the produced foams. In the PUF foams, the urea–urea functional groups’ bidentate and monodentate hydrogen bonding absorbance peaks appear at 1645 and 1668 cm^−1^, respectively.

The bandwidth from isocyanurate groups (PIR) occurs at 1412 cm^−1^ [34], from biuret groups at 1527 cm^−1^ [35], and allophanate groups at 1685 cm^−1^ [36,37]. Carbonyl group compounds are summarized in Table 3.

Monodentate hydrogen bonding peak area intensity is lowest for PUF_85 and highest for PUF_100, similarly changing bidentate hydrogen bonding peak area intensity and biuret groups peak area. According to Baghban et al. [38], the increase in the isocyanate index is associated with the formation of more urea bonds through the reaction of isocyanate groups with OH groups of water [39,40,41].

In the case of PUF foams, the highest amount of urea bonds, including monodentate urea, bidentate urea, and biuret groups, is achieved in PUF100 foam, obtained with an equal proportion of NCO groups to the sum of OH and NH_2_ groups (Figure 3). In foams in this series, the number of urea groupings increases in proportion to the molecular weight of their macromolecules. Due to the much higher reaction rate of urea bond formation than urethane bond formation, fewer urethane bonds capable of forming allophanate groups are formed in PUF foams. In contrast, in PUF_100 foam, allophanates are not formed. In PUF, as INCO increases, the number of isocyanurate rings increases, limiting the mobility of PU macromolecules [34]. As a consequence of the differences in the proportion of urea and urethane groups, there are also differences in the phase separation process in this series of foams.

There are differences in the multiplet signal shape, in the wavenumber range of 1750–1640 cm^−1^ for FTIR spectra between the foams (Figure 3). The differences were related to the change in the intensity of components originating from the stretching vibrations of the carbonyl groups in urethane and urea groups, bound with a hydrogen bond and not bound (Figure 3) [42].

Quantitatively, changes in multiple band components were analyzed, and the procedure for multiple band analysis is described in the paper [43]. Based on the multiple band component analysis results, the hydrogen bonding index (IH), the degree of phase separation (DPS), and the proportion of urea groups (part of urea groups) in the hard phase of the tested foams were calculated. In addition, the proportion of urea groups was calculated as the ratio of the peak area derived from the vibration of the urea groups to the peak area derived from the sum of the peak area of urethane and urea groups.

The results of this analysis are summarized in Table 4.

In PUF_100, the fewest hydrogen bonds are formed, linking the rigid segments, which is probably due to the limited mobility of the macromolecules with the highest molecular weight. This results in the lowest DPS in this foam, although most urea bonds are formed.

#### Cytotoxic Activity of Polyurethane Foam Extracts

The cytotoxicity of polyurethane foams is important for applications where these materials are to be in direct contact with the skin or where the material they are coated with does not protect against their effects. Therefore, studies have been undertaken to assess the effect of various factors in constructing these materials on cytotoxicity. As part of a previous study on open-cell polyurethane foams, a characterization of the structure and properties of these foams was carried out [44]. After this article’s publication, preliminary biological studies were performed to assess their cytotoxicity. The results of these studies are presented in a supplement to this article (Appendix A). A description of the tested open-cell polyurethane foams produced at INCO in the range of 0.6–0.9 is described in the (Appendix A), for a description of the materials used and the methodology of preliminary cytotoxicity tests; see (Appendix A), while the results of these studies are presented in (Appendix A, and Appendix A). These studies concluded that all the foams tested were cytotoxic and that their cytotoxicity independently of INCO was similar. The cytotoxicity of foams made at INCO in the range of 0.85–1.05 was assessed according to the revised methodology and in accordance with ISO 10993-5.

HaCaT keratinocyte cells were subjected to aqueous extracts of three polyurethane foam samples at seven different concentrations (ranging from 1.56% to 100%) with four replicates for each concentration and were exposed for 24 h. The concentration-dependent cytotoxicity curves for each extract are presented in Figure 4 as the mean ± standard deviation (SD) of three independent experiments. The cytotoxicity of all foam extracts remained consistently low, below 10%, for concentrations up to 13%. After that, the cytotoxicity of PUF_85 and PUF_100 foams changed similarly up to a concentration of 25%. Above this concentration, the cytotoxicity of PUF_100 foam increased rapidly up to approximately 50%, the strongest cytotoxicity, while the increase in cytotoxicity for PUF_85 foam was less rapid and increased to approximately 30%. The cytotoxicity of PUF_105 increased at the slowest rate, up to approximately 35%.

These results are in agreement with the IC_50_ values (Table 5), as only for this foam could it be determined, and it was 96.1%. For the remaining foam extracts, it was not possible to determine the IC_50_, as the cytotoxicity was too low, even at the highest concentration tested (100%). Non-toxic concentrations (IC_0_) were as follows: PUF_105 ≤ 25% and PUF_85 ≤ 12.5%; PUF_100 ≤ 6.25%.

Figure 5 displays the alterations in the monolayer of HaCaT cells when exposed to polyurethane foam extracts, while Table 6 showcases the qualitative grading of cytotoxicity based on the ISO 10993-5 standard.

The negative control exhibited regularly shaped cells in a homogeneous monolayer with distinct cell membranes, cytoplasm, and nuclei, and only a few partially detached cells and discrete intracytoplasmic granules were observed. In contrast, exposure to the polyurethane foam extracts destroyed the monolayer, decreased cell count per visual field compared to the negative control, and many rounded cells remained detached or semi-detached from the substrate surface (Figure 5). These microscopic findings were consistent with the cytotoxicity results. According to the qualitative morphological grading of cytotoxicity, severe reactivity was observed only for the positive control (DMSO). At the same time, extract PUF_100 showed moderate reactivity, and two other extracts (PUF_85 and PUF_105) were classified as mild (Table 6). Component B, Ongronat TR 4040, is a blend of MDI isomers and oligomeric MDI. International Fire Code classifies pMDI in the Hazardous Materials Classification Guide as “highly toxic” [45].

## 4. Conclusions

This article describes polyurethane foams for composite matrix synthesized with different isocyanate contents and verifies their effect on the chemical structure and cytotoxicity of the flexible polyurethane foam. The reactivity of the formulation increased with the isocyanate content in the mixture. In-depth FTIR analysis showed differences in obtained monodentate and bidentate urea. There is a possible correlation between the presence of those bonds with the cytotoxicity of the foams.

## Figures and Tables

**Figure 1 polymers-15-02754-f001:**
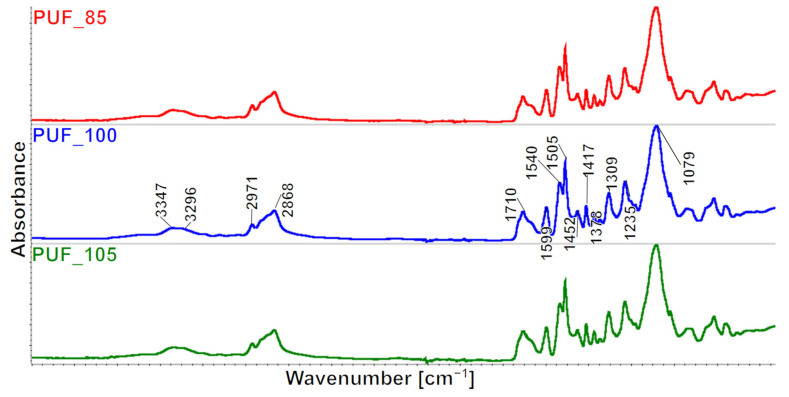
FTIR spectra of the produced foams.

**Figure 2 polymers-15-02754-f002:**
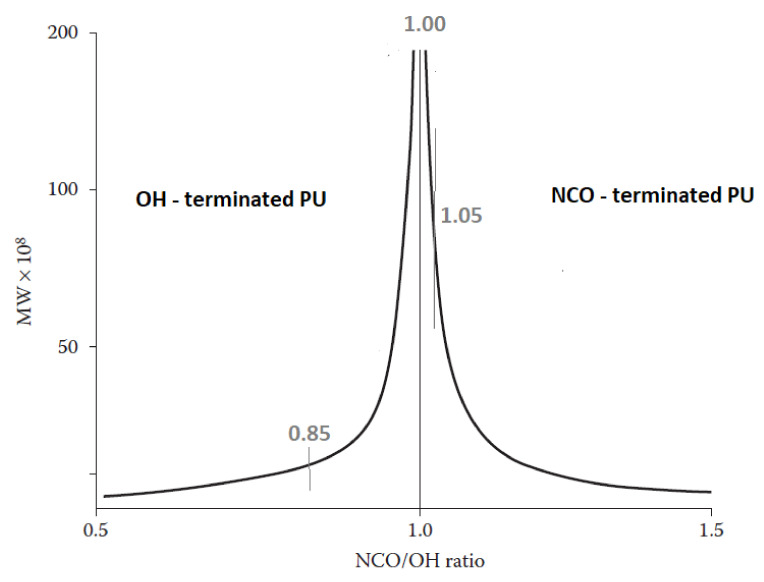
The theoretical relationship between the isocyanate index and molecular weight of polyurethane elastomers is based on [33].

**Figure 3 polymers-15-02754-f003:**
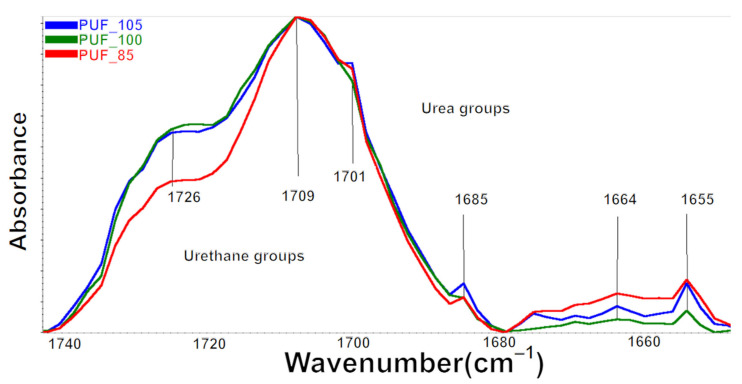
Fragment of the FTIR spectrum from the range of carbonyl groups 1640–1750 cm^−1^.

**Figure 4 polymers-15-02754-f004:**
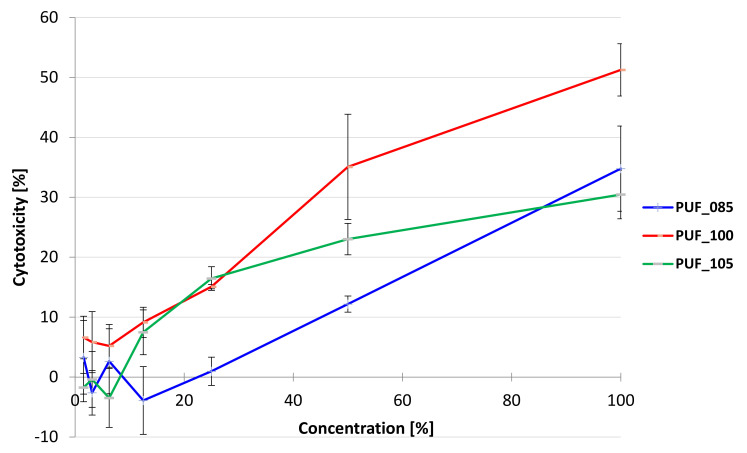
Cytotoxicity of polyurethane foams extracted after 24 h exposition of HaCaT cells (human keratinocyte) in Neutral Red Uptake assay. Each point represents the mean absorbance values of the four replicates from three independent experiments (±standard deviation of the mean—SD).

**Figure 5 polymers-15-02754-f005:**
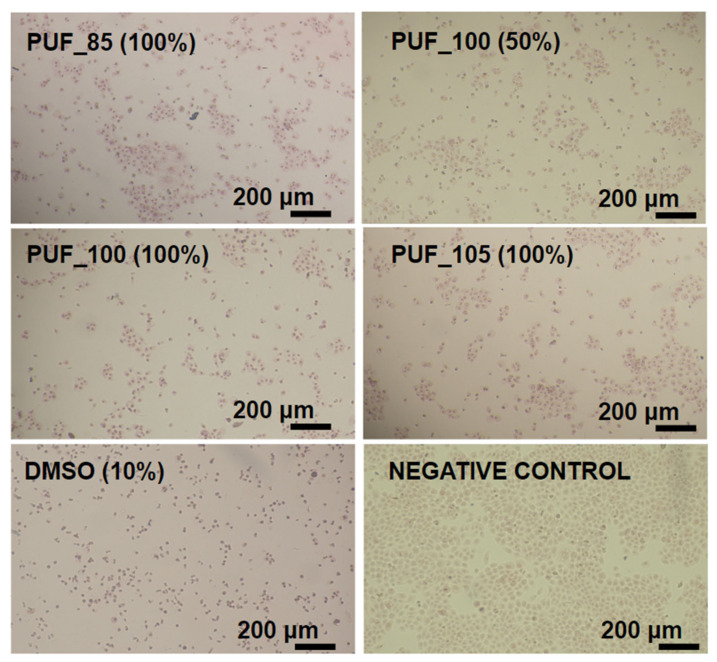
Microphotographs of human keratinocytes HaCaT after 24 h exposition to polyurethane foam extracts after staining with neutral red. Objective 10× (Nikon Ts2, contrast EMBOSS, Tokyo, Japan).

**Table 1 polymers-15-02754-t001:** Isocyanate index of produced PUF foams.

Sample	INCO
PUF_85	85
PUF_100	100
PUF_105	105

**Table 2 polymers-15-02754-t002:** Foam synthesis parameters and apparent density.

Sample	Parameter, s			Apparent Density, kg/m^3^ ± SD
	Start Time	Rise Time	Gel Time	
PUF_85	10	140	270	70.11 ± 1.99
PUF_100	10	132	252	66.00 ± 6.55
PUF_105	10	125	240	62.01 ± 0.49

**Table 3 polymers-15-02754-t003:** Selected groups present in the FTIR spectra of PUFs.

Sample	Monodentate Urea ± SD	Bidentate Urea ± SD	PIR ± SD	Biuret Groups ± SD	Allophanate Groups ± SD
PUF_85	0.42 ± 0.01	0.63 ± 0.01	1.13 ± 0.02	0.88 ± 0.02	0.59 ± 0.02
PUF_100	0.81 ± 0.03	1.09 ± 0.03	1.38 ± 0.04	1.25 ± 0.06	0
PUF_105	0.55 ± 0.02	0.72 ± 0.04	1.43 ± 0.03	1.03 ± 0.04	0.82 ± 0.01

**Table 4 polymers-15-02754-t004:** The analysis results of the hydrogen bonding index (IH) analysis, the degree of phase separation (DPS), and the part of urea groups in the hard phase of foams.

Sample	IH	DPS	Part of Urea Groups [%] ± SD
PUF_85	5.57 ± 0.02	0.848 ± 0.002	56.8 ± 0.3
PUF_100	2.40 ± 0.01	0.706 ± 0.015	57.3 ± 0.2
PUF_105	4.46 ± 0.05	0.815 ± 0.023	52.4 ± 0.8

**Table 5 polymers-15-02754-t005:** IC_50_ values of polyurethane foam extracts after 24 h exposition of HaCaT cells (human keratinocyte) and the average cytotoxicity [%] at the highest tested concentration (± standard deviation of the mean—S.E.M.).

Sample	IC50 [%]	Average Cytotoxicity [%] at Highest Tested Concentration [±SD]
Positive control (DMSO)	8.3	55.2 ± 8.3
PUF_85	nd *	34.8 ± 7.1
PUF_100	96.1	51.3 ± 4.4
PUF_105	nd *	30.5 ± 4.1

* nd—not detected.

**Table 6 polymers-15-02754-t006:** Qualitative morphological grading of cytotoxicity of polyurethane foam extracts (100% concentrations) according to ISO 10993-5 observed in an inverted microscope before adding neutral red.

Sample	Grade	Reactivity	Conditions of All Cultures According to ISO 10993-5
Vehicle control	0	None	Discrete intracytoplasmic granules, no cell lysis, no reduction in cell growth.
Positive control (DMSO)	4	Severe	Nearly complete or complete destruction of the cell layer.
PUF85	2	Mild	No more than 50% of the cells are round, devoid of intracytoplasmic granules; no extensive cell lysis; no more than 50% growth inhibition observed.
PUF100	3	Moderate	No more than 70% of the cell layers contain rounded cells or are lysed; cell layers are not completely destroyed, but more than 50% growth inhibition is observed.
PUF105	2	Mild	No more than 50% of the cells are round, devoid of intracytoplasmic granules; no extensive cell lysis; no more than 50% growth inhibition observed.

## Data Availability

Experimental methods and results are available from the authors.

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
