# Peer review of "Cytotoxic Properties of Polyurethane Foams for Biomedical Applications as a Function of Isocyanate Index"

_polymers, 2023, doi:10.3390/polym15122754_

Round 1
Reviewer 1 Report
1. Equation (1) needs to be made clearer and clarified in the text.
2. Written: “The FTIR spectra of the examined foams exhibit similarity. Specifically, the bands observed in the range of 3600-3400 cm-1 in the FTIR spectra of the foams originate from the stretching vibrations of the -OH groups of the polyol hydroxy groups.”
Further it is written:: “Thus, a broad peak is observable in the spectra within the 3400-3200 cm-1 range, resulting from both asymmetrical and symmetrical stretching vibrations of the -N-H group present in the urethane groups, urea derivatives, and/or the rest of the catalysts.”
Thus, the authors admit a contradiction in the interpretation of the spectra.
In addition, the analysis of the content of hydroxyl groups is not rigorous. A detailed calculation of the conclusions is required: “It can be deduced from the distribution of the multiple bands in this range that PUF_85 contains the most OH groups, which is twice the amount found in PUF_100 and four times more than PUF_105.”
Author Response
Dear Reviewer,
Thank you for the time you put into preparing the review.
- Equation (1) needs to be made clearer and clarified in the text.
We have developed the issues regarding Equation 1; we hope that they will now be clearer for readers.
- Written: “The FTIR spectra of the examined foams exhibit similarity. Specifically, the bands observed in the range of 3600-3400 cm-1 in the FTIR spectra of the foams originate from the stretching vibrations of the -OH groups of the polyol hydroxy groups.”
Further it is written:: “Thus, a broad peak is observable in the spectra within the 3400-3200 cm-1 range, resulting from both asymmetrical and symmetrical stretching vibrations of the -N-H group present in the urethane groups, urea derivatives, and/or the rest of the catalysts.”
Reviewer, we don’t really see a contradiction in those two interpretations.
In addition, the analysis of the content of hydroxyl groups is not rigorous. A detailed calculation of the conclusions is required: “It can be deduced from the distribution of the multiple bands in this range that PUF_85 contains the most OH groups, which is twice the amount found in PUF_100 and four times more than PUF_105.”
The hydroxyl group content is related to the intensity of the peaks. If we use a lower INCO during the synthesis of polyurethanes then there are fewer NCO groups in the reaction mixture. This results in OH groups not having the opportunity to form urethane bonds and remain unbound.
Best regards,
Authors
Reviewer 2 Report
See attachment. I have some difficulties with the scientific conclusions:
1. Why are the results related to the INCO and not of one of the other differences
2. The signal to noise ratio of the cytotoxicity measurements is too high to draw any reliable conclusions in the area of under 20% (figure 4)

Author Response
Dear Reviewer,
Thank you for your suggestions.
Regarding the attachment, we have corrected our paper with your suggestions, such as marking ± as Standard Deviation. About the Reviewer’s comment on the reason for the paper, according to the Supplementary files – The Cytotoxicity results of previous paper weren’t possible to be attached in our study before, we found that mandatory to attach them now.
- Why are the results related to the INCO and not of one of the other differences?
The only thing that was being a variable parameter, was the amount the amount of isocyanate used to create a polyurethane which is closely related to INCO index. In our opinion, it gives the reason to conclude that the amount of isocyanate played a key role here.
- The signal to noise ratio of the cytotoxicity measurements is too high to draw any reliable conclusions in the area of under 20% (figure 4)
Such results are closely related to the nature of these tests. https://www.ncbi.nlm.nih.gov/pmc/articles/PMC2438350/
There is more variability at lower concentrations; the experiments were performed correctly, according to the standard quoted. Three independent experiments were performed; in that case, the standard deviation is always larger because a different population of cells is being tested. In each experiment, the curves followed a similar pattern. We included the background signal in the absorbance measurements - as described in the methodology. The measurement error of this method in our laboratory is about 5-7%
Best Regards
Authors
Round 2
Reviewer 1 Report
Dear Author,
The additional result and discussion help to understand the common readers. Paper can now be accepted.
Reviewer 2 Report
Well done rework